# The Impact of Blenderized Tube Feeding on Gastrointestinal Symptoms, a Scoping Review

Elisabetta Sforza [1,†], Domenico Limongelli [1,†], Valentina Giorgio [1,2,3], Gaia Margiotta [1], Francesco Proli [3], Eliza Maria Kuczynska [2], Chiara Leoni [2,3], Donato Rigante [1,2,3], Ilaria Contaldo [4], Chiara Veredice [4], Emanuele Rinninella [1,5], Antonio Gasbarrini [1,6], Giuseppe Zampino [1,2,3,‡] and Roberta Onesimo [2,3,*,‡]

1   Università Cattolica Sacro Cuore, 00168 Rome, Italy
2   Center for Rare Diseases and Birth Defects, Department of Woman and Child Health and Public Health, Fondazione Policlinico Universitario Agostino Gemelli IRCCS, 00168 Rome, Italy
3   Pediatric Unit, Department of Woman and Child Health and Public Health, Fondazione Policlinico Universitario Agostino Gemelli IRCCS, 00168 Rome, Italy
4   Pediatric Neuropsychiatry, Department of Woman and Child Health and Public Health, Fondazione Policlinico Universitario Agostino Gemelli IRCCS, 00168 Rome, Italy
5   Clinical Nutrition Unit, Department of Medical and Abdominal Surgery and Endocrine-Metabolic Scienze, Fondazione Policlinico Universitario Agostino Gemelli IRCCS, 00168 Rome, Italy
6   Center for Diagnosis and Treatment of Digestive Diseases, CEMAD, Gastroenterology Department, Fondazione Policlinico Universitario Agostino Gemelli IRCCS, 00168 Rome, Italy
*   Correspondence: roberta.onesimo@policlinicogemelli.it
†   Both authors are first authors.
‡   Both authors are last authors.

**Abstract:** Severe gastrointestinal symptoms are one of the main reasons for switching from conventional artificial tube feeding to blenderized tube feeding (BTF). This study aimed to describe and quantify the impact of BTF on gastrointestinal symptoms in children and adults. We analyzed four databases (PubMed, Scopus, Cochrane Library, and Google Scholar). The review was performed following the PRISMA extension for Scoping Reviews checklist. The methodological quality of articles was assessed following the NIH quality assessment tools. The initial search yielded 535 articles and, after removing duplicates and off-topic articles, 12 met the inclusion criteria. All included papers unanimously converged in defining an improvement of gastrointestinal symptoms during blenderized feeding: the eight studies involving pediatric cohorts report a decrease from 30 to over 50% in gagging and retching after commencing BTF. Similar rates are reported for constipation and diarrhea improvement in most critically ill adults. Experimental studies and particularly randomized controlled trials are needed to develop robust evidence on the effectiveness of BTF in gastrointestinal symptom improvement with prolonged follow-up and adequate medical monitoring.

**Keywords:** blenderized tube feeding; enteral nutrition; gastrostomy; children; disabled; review; personalized medicine

## 1. Introduction

Since its introduction in the 1970s, home enteral nutrition (HEN) has been established as a reliable and effective nutritional intervention [1]. Currently, the method of choice for medium- and long-term enteral feeding is the gastrostomy tube (G-tube) [2,3], with a wide range of diets and nutrient preparations suitable for tube feeding [1]. In this context, the European Society for Clinical Nutrition and Metabolism (ESPEN) through the guidelines on HEN recently recommended using standard commercial formulas for enteral tube feeds with the exception of some specific conditions in which blended tube feeds are considered to be the first choice [1]. Specifically, standard commercial formulas refer to standard tube feed made of powdered raw materials [4]. Notably, almost all preparations of tube feedings available on the European market use nutrient isolates (except vegetable oils)

and concentrates in powder instead of natural foods. The nutrients and food isolates (e.g., milk protein) are extracted from foods in these tube feedings but are provided without the natural food matrix [5].

On the contrary, BTF, also referred to as "blenderized formula" or "homemade blended formula" and "pureed by g-tube," consists of whole foods provided through a feeding tube [6,7]. Products based on real foods, such as milk, meat, and vegetables, are also commercially available for enteral tube feeding (ETF) [8]. The number of patients receiving long-term enteral nutrition along with the use of BTF has surged over the last decade. As per the Oley Foundation members' survey in 2017, most of the 216 participants, specifically pediatric (89%) and adult (66%) patients, were consuming BTF partly or totally for their nutritional needs [9]. Although guidelines currently do not recommend BTF as a first choice for tube feeding [1], many families still choose BTF over commercial formulas for several reasons, including severe gastrointestinal (GI) symptoms, intolerance to polymeric enteral formulas, or food allergies and intolerances [10]. Parenteral nutrition can play a role in relieving gastrointestinal signs/symptoms especially in children with severe neurological impairment [11]. However, given its risks and its potential to become inappropriately life sustaining, clinicians need to consider changes in conventional enteral nutrition, including blenderized formula.

This study aims to describe and quantify the impact of BTF on GI symptoms of patients without any age or diagnosis limitations.

## 2. Methods

### 2.1. Search Strategy

Supervised by R.O., E.S. performed a systematic literature search of the following databases: PubMed, Scopus, Cochrane Library, and Google Scholar. Search terms combined text words and Medical Subject Headings (MeSH), as shown in the Supplementary Table S1. Search terms included two components: terms referring to enteral tube feeding and blenderized feeding. No date limit was set. The literature search was conducted in Italy.

### 2.2. Study Eligibility

Following the Preferred Reporting Items for Systematic Reviews and Meta-Analyses Extension for Scoping Reviews (PRISMA-ScR) checklist [12] represented in the Supplementary Table S2, and after removing duplicates, all full-text articles were screened by two independent researchers; any discrepancies were solved in a consensus meeting. The articles were included if (a) they reported GI symptoms in pediatric and adult populations fed via blenderized tubes, (b) were freely available, and (c) written in English. Articles only assessing the nutritional value of formulas and review articles were disregarded.

### 2.3. Data Collection and Assessment

Data were extracted independently from included studies by two authors (E.S. and D.L.) according to a predefined data extraction sheet. Probable disagreements were solved by open discussion between the two authors, and consultation was conducted with a third author (R.O.).

Included studies were assessed independently by two researchers (E.S. and D.L.). To assess the methodological quality of all included articles, we chose three versions of NIH quality assessment tools according to study types. Due to its versatility and completeness, the NIH tool has been increasingly used in the last years for the quality assessment of articles for systematic reviews and meta-analyses. Specifically, we used the Cross-Sectional Study/Observational Cohort/Cross-Sectional Studies tool, Before-After (Pre-Post) Studies With No Control Group Studies tool, and the Controlled Intervention Studies tool [13,14]. After answering each item, two researchers rated the studies' overall quality as having a low-risk bias (good quality), moderate-risk bias (fair quality), or high-risk bias (poor quality).

## 3. Results

### 3.1. Studies Included

The initial literature search yielded 535 potentially relevant articles. After removing duplicates ($n = 150$), 340 "full-text" manuscripts were retrieved. Of them, 12 studies met the inclusion criteria, as shown in Figure 1. The included articles range from 2011 to 2022, spanning an 11-year period.

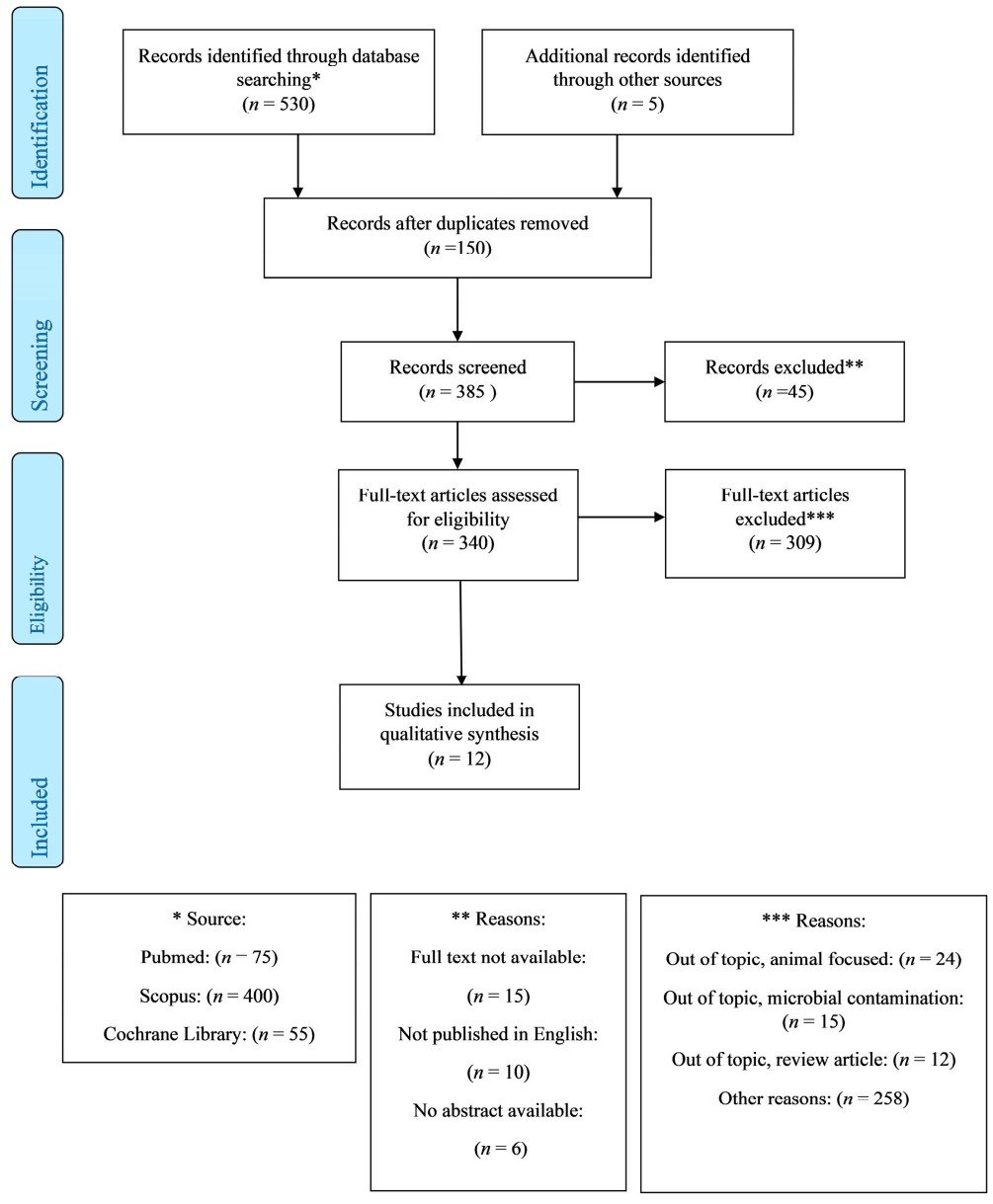

**Figure 1.** PRISMA diagram for article selection and inclusion in the review.

Regarding the study design, of the total studies, 3 were non-experimental cross-sectional studies (surveys) [15–17], 4 were observational retrospective studies [18–21], 5 were semi-experimental longitudinal studies [22–25], and there was only 1 RCT [5]. One study received a financial support from the private sector, specifically from Real Food Blends [25].

### 3.2. Assessment

By focusing on the concepts underlying the questions in the quality assessment tool, no one reached a good overall quality rating among the three cross-sectional studies. Specifically, in studies by Johnson et al., Trollip et al., and Hurt et al. [15–17], potential confounding variables were not measured and statistically adjusted for their impact on the relationship between exposure and outcome. The included observational retrospective studies [18–21] had the lowest risk of bias, although there was a lack of blindness for outcome assessors to the exposure status of participants Table 1. All the semi-experimental studies [22–25] were susceptible to some bias, reducing the quality of the results concerning sample size, follow-up loss rate, blinding of outcome assessors, and statistical analysis (Table 2). Schmidt et al. RCT [5] showed good overall quality. To note, the overall drop-out rate of participants allocated into the intervention and control groups from the study at the endpoint was high (44 to 51%, respectively). Appropriate blinding did not occur as it was not feasible (Tables 3 and 4).

**Table 1.** Quality Assessment of Included Studies using NIH Quality Assessment Tool for Observational Cohort and Cross-Sectional * Studies.

| Study | Q1 | Q2 | Q3 | Q4 | Q5 | Q6 | Q7 | Q8 | Q9 | Q10 | Q11 | Q12 | Q13 | Q14 | Quality (Total Quality Score) |
|---|---|---|---|---|---|---|---|---|---|---|---|---|---|---|---|
| Johnson et al., 2018 * [15] | Y | Y | Y | Y | NR | NO | NO | NO | Y | NA | Y | NO | NA | NO | Fair |
| Trollip et al., 2019 * [16] | Y | Y | Y | Y | NR | NO | NO | NO | Y | NA | Y | NO | NA | NO | Fair |
| Hurt et al., 2015 * [17] | NO | Y | Y | NR | NO | Y | Y | NO | NO | NO | Y | NO | NA | NO | Poor |
| Batsis et al., 2020 [18] | Y | Y | Y | Y | NR | Y | Y | Y | Y | Y | Y | NO | Y | NO | Good |
| Kernizan et al., 2020 [19] | Y | Y | Y | Y | NO | Y | Y | Y | Y | Y | Y | NO | Y | NO | Good |
| Samela et al., 2017 [20] | Y | Y | Y | Y | NO | Y | Y | Y | Y | Y | Y | NO | Y | NO | Good |
| Fabiani et al., 2020 [21] | Y | Y | Y | Y | NO | Y | Y | Y | Y | NO | Y | NO | Y | Y | Good |

CD: cannot determine; NA: not applicable; NIH: National Institutes of Health; NR: not reported; Y: yes. Q1: Was study question or objective clearly stated? Q2: Was the study population clearly specified and defined? Q3: Was the participation rate of eligible persons at least 50%? Q4: Were all the subjects selected or recruited from the same or similar populations (including the same time period)? Were inclusion and exclusion criteria for being in the study prespecified and applied uniformly to all participants? Q5: Was a sample size justification, power description, or variance and effect estimates provided? Q6: For the analyses in this paper, were the exposure(s) of interest measured prior to the outcome(s) being measured? Q7: Was the timeframe sufficient so that one could reasonably expect to see an association between exposure and outcome if it existed? Q8: For exposures that can vary in amount or level, did the study examine different levels of the exposure as related to the outcome (e.g., categories of exposure, or exposure measured as a continuous variable)? Q9: Were the exposure measures (independent variables) clearly defined, valid, reliable, and implemented consistently across all study participants? Q10: Was the exposure(s) assessed more than once over time? Q11: Were the outcome measures (dependent variables) clearly defined, valid, reliable, and implemented consistently across all study participants? Q12: Were the outcome assessors blinded to the exposure status of participants? Q13: Was loss to follow-up after baseline 20% or less? Q14: Were key potential confounding variables measured and adjusted statistically for their impact on the relationship between exposure(s) and outcome(s)?

### 3.3. Findings in the Pediatric Population

Among studies analyzing GI symptoms' prevalence in the pediatric population, the Johnson et al. cross-sectional research reported a lower rate of GI symptoms among BTF users (60% of 217) rather than SCF users (97% of 214). The most frequently reported GI problems in children receiving BTF versus SCF was constipation (18.6% vs. 17.8%) followed by vomiting (13.6% vs. 21%), gas/bloating (11.4% vs. 18.3%), diarrhea (5.4% vs. 11.4%), nausea (3.9% vs. 14.8%), pain (3.9% vs. 11.7%), and fever (1.1% vs. 2.15%) [15].

**Table 2.** Quality Assessment of Included Studies using NIH Quality Assessment Tool for Before-After (Pre-Post) Studies with No Control Group.

| Study | Q1 | Q2 | Q3 | Q4 | Q5 | Q6 | Q7 | Q8 | Q9 | Q10 | Q11 | Q12 | Quality (Total Quality Score) |
|---|---|---|---|---|---|---|---|---|---|---|---|---|---|
| Pentiuk et al., 2011 [22] | Y | Y | Y | Y | NO | Y | Y | NA | Y | NO | Y | NA | Fair |
| Gallagher et al., 2018 [23] | Y | Y | Y | Y | NO | Y | Y | NO | Y | Y | Y | NA | Good |
| Hron et al., 2019 [24] | Y | Y | Y | Y | NO | Y | Y | NO | Y | Y | NO | NA | Fair |
| Spurlock et al., 2022 [25] | Y | Y | Y | Y | NO | Y | Y | NO | NO | Y | Y | NA | Fair |

CD: cannot determine; NA: not applicable; NIH: National Institutes of Health; NR: not reported; Y: yes. Q1:Was the study question or objective clearly stated? Q2: Were eligibility/selection criteria for the study population prespecified and clearly described? Q3: Were the participants in the study representative of those who would be eligible for the test/service/intervention in the general or clinical population of interest? Q4: Were all eligible participants that met the prespecified entry criteria enrolled? Q5: Was the sample size sufficiently large to provide confidence in the findings? Q6: Was the test/service/intervention clearly described and delivered consistently across the study population? Q7: Were the outcome measures prespecified, clearly defined, valid, reliable, and assessed consistently across all study participants? Q8: Were the people assessing the outcomes blinded to the participants' exposures/interventions? Q9: Was the loss to follow-up after baseline 20% or less? Were those lost to follow-up accounted for in the analysis? Q10: Did the statistical methods examine changes in outcome measures from before to after the intervention? Were statistical tests done that provided *p* values for the pre-to-post changes? Q11:Were outcome measures of interest taken multiple times before the intervention and multiple times after the intervention (i.e., did they use an interrupted time-series design)? Q12: If the intervention was conducted at a group level (e.g., a whole hospital, a community, etc.) did the statistical analysis take into account the use of individual-level data to determine effects at the group level?

**Table 3.** Quality Assessment of Relevant Study using NIH Quality Assessment Tool for Controlled Intervention Studies.

| Study | Q1 | Q2 | Q3 | Q4 | Q5 | Q6 | Q7 | Q8 | Q9 | Q10 | Q11 | Q12 | Q13 | Q14 | Quality (Total Quality Score) |
|---|---|---|---|---|---|---|---|---|---|---|---|---|---|---|---|
| Schmidt et al., 2018 [5] | Y | Y | Y | NO | NO | Y | NO | Y | Y | Y | Y | NO | Y | Y | Good |

CD: cannot determine; NA: not applicable; NIH: National Institutes of Health; NR: not reported; Y: yes. Q1: Was the study described as randomized, a randomized trial, a randomized clinical trial, or an RCT? Q2: Was the method of randomization adequate (i.e., use of randomly generated assignment)? Q3: Was the treatment allocation concealed (so that assignments could not be predicted)? Q4: Were study participants and providers blinded to treatment group assignment? Q5: Were the people assessing the outcomes blinded to the participants' group assignments? Q6: Were the groups similar at baseline on important characteristics that could affect outcomes (e.g., demographics, risk factors, co-morbid conditions)? Q7: Was the overall drop-out rate from the study at endpoint 20% or lower of the number allocated to treatment? Q8: Was the differential drop-out rate (between treatment groups) at endpoint 15 percentage points or lower? Q9: Was there high adherence to the intervention protocols for each treatment group? Q10: Were other interventions avoided or similar in the groups (e.g., similar background treatments)? Q11: Were outcomes assessed using valid and reliable measures, implemented consistently across all study participants? Q12: Did the authors report that the sample size was sufficiently large to be able to detect a difference in the main outcome between groups with at least 80% power? Q13: Were outcomes reported or subgroups analyzed prespecified (i.e., identified before analyses were conducted)? Q14: Were all randomized participants analyzed in the group to which they were originally assigned, i.e., did they use an intention-to-treat analysis?

In Trollip et al.'s survey, a marked improvement of upper GI symptoms was reported by most caregivers in children after commencing BTF (*n* = 12). Specifically, the median score obtained through a novel qualitative questionnaire, developed using items from a well-validated scale on feeding assessment in the pediatric age, showed a significant improvement in GI symptoms. Vomiting and nausea frequency changed from 'often' to 'rare', and reflux changed from 'often' to 'rare'. An improvement in aspiration rate was experienced by one-third of the populations analyzed, with only one caregiver reporting a worsening post-BTF initiation. The reported perceived benefit of BTF on clinical outcomes included a positive trend in bowel habits (*n* = 10), specifically in constipation (from 'often' to 'sometimes') and diarrhea rates (from 'rarely' to 'never'). Abdominal pain rates remained primarily unchanged [16].

**Table 4.** Features of Included Studies.

| Study | Population and Sample Size | Outcome Evaluation Method | Blenderized Food—Type | Diet Prescription Guidelines | Follow-Up Period |
|---|---|---|---|---|---|
| Johnson et al., 2018 [15] | 255 children 50.5% (*n* = 173) using CF and 49.5% (*n* = 82) using BTF | Questionnaire | Homemade (59%) or commercially available blenderized | NR | NA |
| Trollip et al., 2019 [16] | 12 children | Questionnaire | Homemade (33%) or a combination of homemade and formula (17.5%) | NR | NA |
| Hurt et al., 2015 [17] | 54 adults 50.5% (*n* = 30) using BTF and 45.5% (*n* = 24) using CF | Self-designed survey | Homemade | NR | NA |
| Batsis et al., 2020 [18] | 23 children | Clinical documentation provided in the medical records | Homemade (65%) or commercially available blenderized (17.5%) or a combination of both (17.5%) | NR | 3 and 6 months |
| Kernizan et al., 2020 [19] | 34 children | Parent report Clinical documentation provided in the medical records | Homemade | Recipes designed by dietitians | NA |
| Samela et al., 2017 [20] | 10 children All transitioned from an elemental formula to real food ingredients formula (TFRF) | The number of defecations and the consistency of each stool | Commercially available blenderized | NR | NA |
| Fabiani et al., 2020 [21] | 250 adults 103 fed blenderized natural enteral feeding and 112 fed commercial formulas. | The number of defecations and the consistency of each stool according to the Bristol Stool Chart (BSC) | Homemade | Simple weight-based equation (25–30 kcal/kg/day) to calculate daily caloric target (Atasever et al., 2018; Taylor et al., 2016) | 8 days |
| Pentiuk et al., 2011 [22] | 33 children | Survey | Homemade | Food Processor Program (ESHA Research, Salem, OR) | 2 months |
| Gallagher et al., 2018 [23] | 20 children | Questionnaire | Homemade | Canada's Food Guide for Healthy Eating. VitamixR 7500 G-Series blender | 6 months |
| Hron et al., 2019 [24] | 70 children | Pediatric Gastroesophageal Reflux Disease Symptom and Quality-of-Life Questionnaire (PGSQ); (L) PedsQL Gastrointestinal Symptoms Scale (GI-PedsQL). | Homemade (67%) or commercially available blenderized | Cronometer, a web-based nutrient database (Revelstoke, British Columbia) | 1 year |
| Spurlok et al., 2022 [25] | 14 adults first 2 weeks in CF, next 3 weeks partial BTF, next full BTF | Questionnaire | Commercially available blenderized | NR | 6 weeks |
| Schmidt et al., 2019 [5] | 118 adults 50% using commercially available product based on real foods and 50% using standard tube feed | The number of defecations and the consistency of each stool according to the Bristol Stool Chart (BSC) | Commercially available blenderized | FAO Expert Consultation on Energy and Protein Requirements (1985) | 1 month |

NA: not applicable, NR: not reported.

Data on GI symptoms retrospectively collected by Batsis et al. on 23 children switching from SCM to BTF highlighted an overall improvement. The majority of the included population (*n* = 21/23) while on standard enteral formulas complained about upper GI symptoms, namely gagging (39%), emesis (48%), and chronic cough with concern for aspiration (4%). Ninety-five percent of them experienced an improvement in the reported symptoms over a three-month period. BTFs did not decrease the constipation rate in the patient previously suffering from it, and new onset mild constipation was reported in 21% (*n* = 5) of patients [18].

Similar data were reported by Kernizan et al. in 35 highly complex patients, almost all (*n* = 33) suffering from GI symptoms before switching to partial or full BTF. Sixty percent of them (*n* = 21/35) experienced a gradual improvement of symptoms during the follow-up visits. The most commonly improved symptoms were those related to gastroesophageal reflux disease (GERD). Two patients had worsening of constipation and GERD, respectively [19].

Furthermore, Samela et al. reported a reduction in lower GI symptoms in 10 patients suffering from intestinal failure to absorb macro- and micronutrients, transitioning from SCF to tube feeding formula with real food ingredients (TFRF). Specifically, stooling patterns (consistency or volume and number in 24 h) improved in the majority of cases (90%). TFRF resulted in being well tolerated in children with 30–40 cm of small bowel, an intact ileocecal valve, and at least two-thirds of their colons in continuity [20].

Petniuk et al. reported that fifty-two percent (*n* = 17/33) of children with fundoplicatio experienced an extensive decrease (76% to 100%) in gagging and retching after two months on BTF. No parents reported that their child's GI symptoms worsened after starting BTF [22].

In a more extended monitoring period of six months for 33 patients, Gallagher et al. recorded a decrease in gagging and retching from 82% pre-BTF to 47% post-BTF. Between enrolment and study exit, stool frequency of more than one/day slightly decreased (from 100% to 94%), while stool consistency did not significantly change [23].

In a larger cohort study of Hron et al. on 70 children, participants receiving blenderized diets compared with those receiving SCF showed fewer GERD symptoms assessed by the Pediatric Gastroesophageal Symptom and the Quality-of-Life Questionnaire (PGSQ). Furthermore, participants on blenderized diets indicated an overall improved gastrointestinal function through Pediatric Quality-of-Life Inventory Gastrointestinal Symptoms (PedsQL). Specifically, less nausea and vomiting (64.0 $\pm$ 22.6 vs. 49.0 $\pm$ 37.9, *p* = 0.02), less abdominal pain (65.0 $\pm$ 26.8 vs. 56.4 $\pm$ 33.9, *p* = 0.04), abdominal upset (71.1 $\pm$ 26.0 vs. 58.9 $\pm$ 32.7, *p* = 0.02), less diarrhea (87.9 $\pm$ 15.5 vs. 73.6 $\pm$ 26.3, *p* = 0.004), and less worry about stool (91.5 $\pm$ 12.8 vs. 81.4 $\pm$ 30.0, *p* = 0.05) were reported [24].

*3.4. Findings in the Adult Population*

In Hurt et al.'s survey, among 54 adults using either BTF (*n* = 30) or CF (*n* = 24), nausea/vomiting was scarcely reported (13%) in both groups. On the contrary, diarrhea was experienced by 21% of the 24 patients using SCF and by 16% of those using BTF. Constipation was reported by 6% and 3%, respectively, in the BTF and SCF groups [17].

Over an 8-day-observation window, Fabiani et al. found that roughly half of the 112 critically ill patients (due to cardiac surgery) fed with SCF developed diarrhea, while this symptom occurred in less than one-third of the 103 patients fed by BTF [21].

In a cohort of adults with head and neck cancer, Spurlock et al. found that all GI symptoms improved after switching from SCF to BTF. GI symptoms decreased, particularly vomiting (31.3% to 12.5%), constipation (31.3% to 12.5%), gas/bloating (50% to 18.8%), nausea (62.5% to 12.5%), and diarrhea (37.5% to 0%) [25].

Lastly, Schmidt et al. in their RCT provided that in critically ill neurological patients BTF may considerably reduce the number of watery stools and diarrhea, over a 24-day-observation of complete enteral nutrition, when compared to fiber-based SCF [5].

## 4. Discussion

The sentence "What is old is new again" is particularly true when considering BTF. While is it ancient like old Egyptians, the popularity and necessity of this type of nutrition reduced with the emergence of commercial formulas in the middle 20th century. As technology advanced over the 1960s and 1970s, SCF (for definition, sterile products) began to replace home-prepared food because of their known nutrient composition and lack of possible microbial contamination [8]. However, lately, the interest in these individualized formulas has increased, especially as a request made by caregivers (a way of feeding perceived more "normal" by the family) [10].

Thirty years ago, the estimated annual prevalence of HEN in the USA was 415 per million people [26]. The practice of HEN has faced extensive growth over time, especially in the pediatric population, where the estimated overall prevalence is 3.47 per 100,000 inhabitants from 0 to 18 years of age [27]. This increase is driven by the rising prevalence of feeding and swallowing difficulties connected to improved survival rates of children with complex disabilities and rare genetic conditions [28–32]. In addition to SCF intolerance, one reason of the emerging interest and use of BFT is the inability to obtain commercial formulas in some peculiar settings [33]. To date, in some developing economies, such as Iran, for most hospitals, the traditional blended formulas remain the most widely used option due to higher affordability [34]. Conversely, in the Medicare and Medicaid context, food-based products are a second choice, preferred in case of allergy or intolerance to semi-synthetic formulas [26]. Although SCF guarantees an adequate supply of nutrients, with a low risk of contamination and device obstruction, intolerance has been reported [10]. The improvement of this latter concern is reported in multiple studies both in pediatric and adult populations [8,9,17].

Many aspects have contributed to the re-emergence of BTF. Paramount, for example, is the improvement in feeding tolerance (reduction in reflux, retching/gagging, constipation). This, indeed, greatly improved psychosocial aspects related to feeding, like normalizing mealtimes, allowing patients to participate in food preparation, and allowing caregivers to fulfill the fundamental role of feeding their child [8]. The BLEND study by Pentiuk et al. found a clear improvement in the QoL of families in which BTF was adopted [22].

However, some critical points are to be considered by clinicians before initiating BTF. The patient should be medically stable on a home enteral nutrition regimen, tolerate bolus feedings, and have access to the necessary equipment to prepare and store food. The gastrostomy site should be mature and well-maintained, and the gastrostomy tube should be ≥14 French to reduce the risk of tube occlusion [8]. Nonetheless, there are some concerns about BTF. Food-borne illness, related to the preparation of a "whole food formula", is a possible "side effect" that commonly leads clinicians to prefer commercial formula and represents a reason of concern for patients/caregivers, particularly for critically ill or immunocompromised patients, such as neonates [35].

Regardless, no available studies demonstrate a connection between higher levels of bacterial contamination and increased infection rates in BTF [36]. Additionally, a 2020 study by Milton et al. shows that, when a correct way of preparation of BTF is applied, 88% of samples meet the criteria for safe food consumption [37].

With regard to contamination concerns, many authors suggest the use of accepted methods of safe food handling [37], adequate hygiene measures, and the use of comprehensive guidelines for preparation, storage, and transportation of BTF [38,39].

A review based on the adult population showed that BTFs are inappropriate for use in medically complex patients or those at risk for malnutrition, since BTF seems to be associated with lower nutrient adequacy, possibly leading to a decline in weight status, BMI, and upper arm circumference [40]. This aspect could be of concern in more fragile pediatric patients.

Overall volume tolerance should also be factored in. Given the patient's level of volume sensitivity, the patient could not meet the daily caloric and nutrient intake needed with BTF alone, which is crucial in children's growth [8]. The BLEND3 study found that

children needed 1.5-fold calories when on BTF in comparison to SCF to sustain growth. It is still controversial why a caloric increase is necessary for BTF. Possible explanations include differences in thermic effects of feeding, miscalculation of the caloric value of foods, or changes in food digestion on BTF. However, further investigation is needed to clarify this point [36].

The most relevant evidence emerging from this research study is that all included papers unanimously converge in defining an improvement in both upper and lower GI symptoms during BTF both in the pediatric and adult populations regardless of their medical conditions. GI symptoms, including diarrhea and abdominal distension, frequently occur in patients receiving EN, and diverse causes, such as antibiotics, infections, or even enteral nutrition, may contribute [41].

All included studies showed comparable results, indicating an improvement of HEN tolerance between patients using BTF. Notwithstanding, experimental studies, particularly RCTs, are lacking. Adequate methodological quality is only sometimes achieved in the included studies in this review. In fact, dietetic prescription, as well as concomitant possible antibiotic administration, are not always described owing to observational biases. To note, blinding is more difficult to achieve in studies on feeds as less feasible. Additionally, due to small cohorts, statistical power is lacking.

Despite all the limitations, this review highlights the possible improvement of BTF on clinical outcomes. It fits into a stream of studies covering most aspects of HEN, including future strategies to improve environmental sustainability of HEN [42]. Results of studies on nutritional value, quality of life, and microbial contamination of BTF are not always consistent, as they do not always show that BTF is more beneficial than SCF.

Notably, BTF is not a good alternative in patients requiring jejunostomy in order to prevent metabolic complications. Since this feeding route requires the administration of feeds through a feeding pump, BTF is not recommended [1,43–45].

To conclude, there is a strong consensus agreement on most the appropriate HEN formula among experts. ESPEN guideline recommends the use of SCF as the first choice, unless there is the presence of a specific justification for BTF [1]. The absence of standardized BTF formulas may potentially increase the risk of malnutrition due to deficiencies of micronutrients [9,44] that are fundamental for effective metabolism and biochemical processes [46,47]. Hence, the importance of an appropriate interdisciplinary management of BTF, especially by an expert dietitian. A 3-day food diary may be filled in to estimate the adequate supply of macro- and micronutrients and to prevent weight loss.

## 5. Conclusions

Empirically, BTF is sometimes preferred to the more conventional CTF because of the emergence of GI symptoms during enteral nutrition with CTF. Although several studies conducted on adults and pediatrics report an improvement in GI symptoms' frequency during BTF in comparison with CTF, only a few studies report with a high degree of methodological quality. Considering these findings, experimental research is needed to develop most robust evidence on this topic that is gaining increasing consideration among caregivers and patients.

## 6. Future Directions

Despite the difficulties with blinding and conducting RCTs in feeding studies, more studies are needed to evaluate the effectiveness of blenderized tube feedings with prolonged follow-up period and adequate medical monitoring to ensure optimal delivery and nutritional standards. In the future, it will be interesting to consider the economic and eco-sustainable impact of BTF in addition to the evaluation of clinical competence.

**Supplementary Materials:** The following supporting information can be downloaded at: https://www.mdpi.com/article/10.3390/app13042173/s1, Table S1: Methodology of search for articles evaluated in this Review; Table S2: Preferred Reporting Items for Systematic reviews and Meta-Analyses extension for Scoping Reviews (PRISMA-ScR) Checklist, From [12].

**Author Contributions:** Study concept and design: R.O., E.S., D.L. and G.Z. Acquisition of data: E.S. and R.O. Analysis and interpretation of data: E.S. and D.L. Drafting the manuscript: E.S. and R.O. Critical revision of the manuscript for important intellectual content: D.R., V.G., E.R. and G.Z. Conceptualization and editing of the manuscript: R.O., E.S., D.L., V.G., G.M., F.P., E.M.K., C.L., D.R., I.C., C.V., E.R., A.G. and G.Z. All authors have read and agreed to the published version of the manuscript.

**Funding:** This research received no external funding.

**Institutional Review Board Statement:** Not applicable.

**Informed Consent Statement:** Not applicable.

**Data Availability Statement:** Not applicable.

**Conflicts of Interest:** The authors declare no competing interest.

## Abbreviations

| | |
|---|---|
| BTF | blenderized tube feeding |
| ESPEN | European Society for Clinical Nutrition and Metabolism |
| ETF | enteral tube feeding |
| G-tube | gastrostomy tube |
| HEN | home enteral nutrition |
| SCF | standard commercial formula |

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
