# Peer review of "The Impact of Blenderized Tube Feeding on Gastrointestinal Symptoms, a Scoping Review"

_applsci, doi:10.3390/app13042173_

Round 1

Reviewer 1 Report

Overall a well written scoping reviewing on the effect of blended tube feeds on GI symptoms for children and adults

A few points:

- There have been a few scoping reviews published on BTF and GI symptoms in recent years, what is new or unique about this review?

- Why were studies after 2020 not included? There appear to be a few papers which have been missed which fit your inclusion criteria

- The assessment section is written a little awkwardly. May need some rewording for better clarity/sentence flow

- A few sentences need to be checked for correct grammar, for example "In a review based on the adult population, BTF results inappropriate for use in patients that are medically

complex or at risk for malnutrition . . .". This should say BTF results are inappropriate . . ."

- Figure 2 is not needed

Author Response

Dear Reviewers,

we want to thank you again for reviewing our manuscript entitled “RTIFICIAL NUTRITION CHALLENGING: THE IMPACT OF BLENDERIZED TUBE FEEDING ON GASTROINTESTINAL SYMPTOMS, A SCOPING REVIEW” submitted to Applied Sciences.

Please find below a detailed point-by-point response to all comments (reviewers’ comments in black, our replies in blue).

REV #1

Overall a well written scoping reviewing on the effect of blended tube feeds on GI symptoms for children and adults

AA: We thank the Reviewer for the positive comments.

A few points:

- There have been a few scoping reviews published on BTF and GI symptoms in recent years, what is new or unique about this review? 

AA: We think the strength lies in evaluating the same intervention in both children and adults with different outcomes, as outlined in the introduction paragraph.  

- Why were studies after 2020 not included? There appear to be a few papers which have been missed which fit your inclusion criteria

AA: It is actually a typo, we do apologyse. Studies after 2020 have already been included e.g. Spurlock et al. 2022.

- The assessment section is written a little awkwardly. May need some rewording for better clarity/sentence flow

AA: We have made some slight adjustments to the wording for better clarity/sentence flow, as suggested.

- A few sentences need to be checked for correct grammar, for example "In a review based on the adult population, BTF results inappropriate for use in patients that are medically complex or at risk for malnutrition . . .". This should say BTF results are inappropriate . . ."

AA: We have checked for correct grammar, as requested.

- Figure 2 is not needed

AA: We have deleted Figure 2, as suggested.

We would like to know if it is possible to keep it in the supplementary material.

REV #2

The work was conducted with a good method, I recommend a revision of the English language and an adaptation of the form of the bibliography

AA: Dear Reviewer, thanks for considering our paper of interest. All adjustment required have been provided in the new version of the manuscript.

REV #3

This is an interesting and topical scoping review paper with a good logical review of the relevant literature. Appropriate methodology is employed, and explanations given for study design. Throughout the paper, the current body of literature is appropriately cited however some parts of the paper need references/further references. Results, discussion (particularly second half of it) and conclusion require many clarifications and re-wording, for strength of the paper and ease of understanding for the reader. The tables are useful. Advise caution around wording of assessment of quality of other authors papers, especially where those authors state their limitations, and it is clear why things like blinding are extremely difficult in feeding research. This review, once manuscript is strengthened, could add significant value to the field.

AA: We thank the Reviewer for the insights and helpful comments.

Commented [A1]: Title unclear. Suggest THE IMPACT OF BLENDERIZED TUBE FEEDING ON GASTROINTESTINAL SYMPTOMS, A SCOPING REVIEW

AA: We have changed the title.

Commented [A2]: Below this is abbreviated as 'Blenderized tube feeds' which is different. Use one consistently

AA: We now have used BTF consistently.

[A3]: Unclear sentence structure.

AA: We have modified the sentence structure.

[A4]: Clarify if you are referring to quality of just the included papers

AA: We have clarified, we were referring to the included papers.

[A5]: Unclear. Do you mean those who use BTF? Or prefer to use it?

AA: We meant ‘use’. We have modified the sentence.

[A6]: Clarify which associations and use wording like 'currently don't recommend' as opposed to 'cannot'

AA: We have made the clarification and made the re-wording.

[A7]: This is very unclear. Describe using more words.

AA: We made the sentence clearer.

[A8]: Blinding was not possible in the Schmidt study, likely due to fact BTF looks so obviously different to commercial formula. Suggest re-word to blinding did not occur as it was not feasible. Overall this manuscript would benefit from a couple of lines mentioning how blinding is difficult in feed studies and why.

AA: We have mentioned the difficulty in blinding in fed studies.

[A9]: Hurt et al define their population as over 18, under follow up care and on commercial feed. Please consider if the answer here should be a Y

AA: We have made the change.

[A10]: Consider reword to 'relevant study' as there is only one

AA: We have considered the re-wording.

[A11]: Need abbreviations below table to state what 'nr' means

AA: We have added the abbreviations.

[A12]: Clarify it is commencing children on BTFs

AA: We have made the clarification.

[A13]: Paper will be easier to read if you use 'often' 'never' etc

AA: We have made the suggested changes.

[A14]: Unclear what is meant by referring. Do you mean reporting or recording?

AA: We meant reporting. We have made the correction.

[A15]: Unclear. Do you mean improvement?

AA: We meant improvement. WE have made the correction.

[A16]: It would be easier to read if a consistent method of referring to commercial feeds was used. Earlier they were called CF

AA: We have consistently referred to standard commercial formulas (SCF) referring to conventional commercial formulas, through the manuscript, as suggested.

[A17]: Clarify if this was after using full or partial BTF if this is known

AA: We have clarified this was after using both partial and full BTF.

[A18]: Use consistent terminology

AA: We have implemented use of consisted terminology.

[A19]: Clarify with some commentary, whether these interesting findings are deemed significant (if statistical tests were carried out on this)

AA: Statistical tests on this was not carried out, significancy was not calculated.

[A20]: This sentence is very unclear. Please re write.

AA: We have re-written the sentence for better clarity.

[A21]: What is meant by ease? Ease of use? Compared to something else?

AA: We have modified the sentence including the term ‘ease’.

[A22]: Clarify here if these feeds are sterile. Suggest you read and consider referencing this paper also by Ojo, 2020 https://doi.org/10.3390/ijerph17249563

AA: We have made the reference to Ojo et al. in the section regarding contamination concerns.

[A23]: Add reference. Suggest consider Bennett and or Ojo 2020

AA: We have made the reference to Ojo et al. in the section regarding contamination concerns.

[A24]: Clarify if this is children

AA: We have clarified the reference to children.

[A25]: This is most unclear what prevalence you are referring to

AA:  We have specified the prevalence we were referring (feeding and swallowing difficulties)

[A26]: This sentence is unclear. Suggest re writing it and breaking it down into two sentences to clearly get your point across

AA: We have re-written the sentence for better clarity.

[A27]: Suggest - 'second option type for formula feeding'

AA: We have re-written the sentence for better clarity.

[A28]: What is your reference for this reporting?

AA: We have added the reference for this reporting.

[A29]: It would be good to include an image of an adult using BTF also. Suggest title re-word to Paediatric patient using homemade BTF (or similar)

AA: We have removed the image as suggested by second Reviewer, possibly maintaining it as supplementary material. We unfortunately are not able to provide an image of an adult using BTF.

[A30]: Consider removal of perceived

AA: We have removed perceived.

[A31]: Caregivers may not be parents

AA: We have made the change.

[A32]: Do you mean BTF is used?

AA: We meant ‘used’.

[A33]: Clarify who you are saying needs to consider these points. Clinician? Caregiver?

AA: We have clarified ‘clinicians’ needs to consider these points.

[A34]: Define these earlier as advised, then remove this here

AA: We have made the changes.

[A35]: Is this BTF? Clarify

AA: We clarified we were referring to BTF.

[A36]: This sentence structure is very unclear, it needs to be re-worded

AA: We have implemented the re-wording.

[A37]: This is very unclear

AA: The have clarified the sentence.

[A38]: Clarify if this refers to the included studies only

AA: We have clarified this refers to the included studies only.

[A39]: This is an important part of the paper and it is not extensive enough or clear enough. What stream of studies do you mean? Are you referring to papers on BTF only? Are you saying results do not always show that BTF is more beneficial/advantageous? Also the paper needs some commentary on why constipation may be a factor that does not seem to be improved, or can be slightly worse (perhaps not clinically or statistically significant though?) with BTF use.

AA: We have implemented some clarification and extension.

[A40]: The section on contamination seems out of place here. Consider moving and consolidating with other mentions of this topic. Also highly recommend you reference ESPEN and or ASPEN guidelines where they exist, on matters such as who may be suitable to get prescribed BTF.

AA: We have moved the section. We have added ESPEN and ASPEN references.

[A41]: Unclear what you mean by feeds for HEN. Clarify if you mean most appropriate formula/feed types AA: We have clarified the meaning, as referencing to the most appropriate formula.

[A42]: Clarify if you mean standardized BTF here

AA: We have clarified we were referencing to standardized BTF.

[A43]: Clarify this is enteral not inclusive of parenteral. Overall the sentence is not very clear and needs more words

AA: We have reformulated the sentence for better clarity.

[A44]: Is this improvement in adults and or paediatrics?

AA: We were referring to both adults and pediatrics.

[A45]: Clarify if you mean reliability based on the tools you used to assess this, or some other factor

AA: We were referring to methodological quality.

[A46]: This section will benefit by you adding more to the manuscript about difficulties with blinding and conducting RCTs in feeding studies

AA: We have pointed out the difficulty in blinding in feeding studies.

[A47]: This is the first mention of sustainability and therefore seems a bit out of context. Suggest adding a bit more on this somewhere in paper if you intend to mention it here

AA: We have added a bit more with reference [42] in the discussion section.

Reviewer 2 Report

the work was conducted with a good method, I recommend a revision of the English language and an adaptation of the form of the bibliography

Author Response

(The authors gave the same response as above.)

Reviewer 3 Report

This is an interesting and topical scoping review paper with a good logical review of the relevant literature. Appropriate methodology is employed, and explanations given for study design. Throughout the paper, the current body of literature is appropriately cited however some parts of the paper need references/further references. Results, discussion (particularly second half of it) and conclusion require many clarifications and re-wording, for strength of the paper and ease of understanding for the reader. The tables are useful. Advise caution around wording of assessment of quality of other authors papers, especially where those authors state their limitations, and it is clear why things like blinding are extremely difficult in feeding research. This review, once manuscript is strengthened, could add significant value to the field.

Please see extensive comments, and highlighted mis-spellings, in attached file.

Author Response

(The authors gave the same response as above.)

Round 2

Reviewer 3 Report

Paper reads better. good luck